# Genome-Wide Association Study of Growth Traits in a Four-Way Crossbred Pig Population

**DOI:** 10.3390/genes13111990

**Published:** 2022-10-31

**Authors:** Huiyu Wang, Xiaoyi Wang, Mingli Li, Hao Sun, Qiang Chen, Dawei Yan, Xinxing Dong, Yuchun Pan, Shaoxiong Lu

**Affiliations:** 1Faculty of Animal Science and Technology, Yunnan Agricultural University, Kunming 650201, China; 2Faculty of Animal Science, Xichang University, Xichang 615000, China; 3Faculty of Agriculture and Biology, Shanghai Jiao Tong University, Shanghai 200240, China; 4Faculty of Animal Science, Zhejiang University, Hangzhou 310058, China

**Keywords:** genome-wide association study, pigs, growth traits, SLAF-seq, SNPs, genes

## Abstract

Growth traits are crucial economic traits in the commercial pig industry and have a substantial impact on pig production. However, the genetic mechanism of growth traits is not very clear. In this study, we performed a genome-wide association study (GWAS) based on the specific-locus amplified fragment sequencing (SLAF-seq) to analyze ten growth traits on 223 four-way intercross pigs. A total of 227,921 highly consistent single nucleotide polymorphisms (SNPs) uniformly dispersed throughout the entire genome were used to conduct GWAS. A total of 53 SNPs were identified for ten growth traits using the mixed linear model (MLM), of which 18 SNPs were located in previously reported quantitative trait loci (QTL) regions. Two novel QTLs on SSC4 and SSC7 were related to average daily gain from 30 to 60 kg (ADG30–60) and body length (BL), respectively. Furthermore, 13 candidate genes (*ATP5O*, *GHRHR*, *TRIM55*, *EIF2AK1*, *PLEKHA1*, *BRAP*, *COL11A2*, *HMGA1*, *NHLRC1*, *SGSM1*, *NFATC2*, *MAML1*, and *PSD3*) were found to be associated with growth traits in pigs. The GWAS findings will enhance our comprehension of the genetic architecture of growth traits. We suggested that these detected SNPs and corresponding candidate genes might provide a biological foundation for improving the growth and production performance of pigs in swine breeding.

## 1. Introduction

Pork is a popular form of animal protein and is now one of the main sources of human dietary protein. Growth and body size traits, including live backfat thickness (LBT), average daily gain (ADG), body length (BL), body height (BH), back height (BAH), chest circumference (CC), chest depth (CD), and rump circumference (RC), are vital economic traits in the swine industry and have a crucial effect on pig production [1,2,3]. Among them, LBT and ADG in different stages are vital indicators of the growth rate of pigs due to their significant impact on production efficiency [1]. Furthermore, body size traits such as BL, BH, BAH, CC, CD, and RC are closely related to body growth and pork production, and feed efficiency. The body character index is frequently employed in pig breeding as the most direct production indicator of pigs [3]. Both genetic and non-genetic effects, including pig breed, feeding behavior, and nutrition level, influence growth traits of pigs [4,5,6]. In the past few decades, pig growth performance has been improved by traditional breeding methods. Growth traits, nevertheless, are intricate quantitative traits controlled by a few major genes and numerous minor genes and their genetic architecture is complex. Therefore, the effect of improvement growth traits through conventional breeding is limited. Due to the rapid development of molecular markers and the completion of the pig genome sequence, molecular breeding has become an effective way to improve growth traits. To date, a total of 2597 quantitative trait loci (QTL) associated with growth traits have been added to the pig QTL database (http://www.animalgenome.org/cgi-bin/QTLdb/index (accessed on 25 April 2022)). These findings have considerably improved our knowledge of the genetic architecture of pig growth traits. However, poor resolution in QTL mapping experiments and the complicated genetic architecture of many QTLs result in an unavoidable challenge for identifying causative mutations [7].

Genome-wide association study (GWAS) has been increasingly used to identify variants and functional genes associated with traits of interest with high resolution with the development of high-density single nucleotide polymorphism (SNP) arrays and the reduction of high-density SNP analysis costs. GWAS based on the porcine SNP array for growth-related traits of pigs has identified many QTLs and candidate genes in recent years [8,9,10,11,12,13,14,15,16,17,18,19,20,21]. In a previous study, 2084 Duroc pigs were genotyped with a 50 K SNP array, and 21 genes were identified as candidate genes for growth traits through GWAS [22]. Using the Illumina PorcineSNP60 BeadChip, Long et al. [23] identified many novel SNPs and two positional candidate genes (*CACNA1E* and *ACBD6*) associated with BFT and ADG, and Cai et al. [24] detected 8 candidate genes (*MRAP2*, *LEPROT*, *PMAIP1*, *BMP2*, *ELFN1*, etc.) significantly related to ADG for three Danbred pig breeds, Duroc, Landrace, and Yorkshire. Liu et al. [3] found 11 candidate genes (*MAPK4*, *HMGA1*, etc.) associated with body size traits in pigs using GWAS based on Illumina Porcine 80K SNP chips. However, GWAS based on the porcine SNP array could only detect a small number of known SNPs, and the detected markers were not evenly distributed throughout the genome. Furthermore, GWAS based on whole-genome sequencing (WGS) is another genotyping method that has been used in recent years. Nevertheless, GWAS for *Sus scrofa* (Sscrofa) with large populations based on WGS is still prohibitively expensive at present. Therefore, specific-locus amplified fragment sequencing (SLAF-seq), a reduced representation sequencing technology was developed, which could create large-scale SNP data quickly, reliably, efficiently, and affordably [25]. SLAF-seq technology based on high-throughput sequencing could generate millions of high-density SNP loci covering the whole genome and could detect novel SNP loci in unknown mutations compared with SNP arrays. Abundant SNPs and candidate genes have been identified using SLAF-seq-based GWAS for various economic traits in different livestock [26,27,28,29,30,31]. Abundant novel mutation sites were also effectively detected using SLAF-seq for pig genotyping [32,33]. Furthermore, we also found two regions on SSC7 and SSC9 and eight potential candidate genes related to porcine fatness-related traits using GWAS based on SLAF-seq in our previous research [34].

To generate more genetic variation to study the complex genetic structure of important economic traits, including growth traits, a four-way crossbred pig population was established in the current study, in which Landrace, Yorkshire, and Duroc were used as hybrid males and Saba pigs as hybrid females. Western commercial pigs are significantly different from Chinese native pigs in terms of growth performance and physical characteristics. Large White, Landrace, and Duroc with fast growth and a high lean percentage are typical Western commercial breeds, which are widely dispersed throughout the world. However, the Saba pig is a fat-type pig breed, which is an invaluable Chinese genetic resource and is distributed in Yunnan Province, China [35]. Due to excessive fat deposition in Saba pigs, the growth rate and the feed conversion rate are lower than those of Western lean pig breeds. Thus, the offspring showed great differences in growth traits using Chinese Saba pigs and Western pig breeds as hybrid parents.

In this study, ten growth traits, including live backfat thickness (LBT), average daily gain from 30 to 60 kg (ADG30–60), average daily gain from 60 to 100 kg (ADG60–100), average daily gain from 30 to 100 kg (ADG30–100), body length (BL), body height (BH), back height (BAH), chest circumference (CC), chest depth (CD), and rump circumference (RC), were examined on 223 four-way crossbred pigs raised in the same environmental conditions. GWAS was then conducted based on SLAF-seq technology to identify significant SNPs associated with these traits. To our knowledge, there is the first report about genome-wide studies for growth-related traits in pigs using the SLAF-seq technology. The outcomes offer a foundation for pig breeding and the improvement of growth traits using molecular markers.

## 2. Materials and Methods

### 2.1. Ethics Statement

All of the animals employed in this research were treated and used following the standards established by the Ministry of Agriculture and Rural Affairs of China for the care and use of experimental animals. The ethics committee of Yunnan Agricultural University (YNAU, Kunming, China) approved the entire research.

### 2.2. Animals, Phenotypic Collection and Statistical Analysis

A four-way crossbred pig population was established as described previously [34]. In short, 7 hybrid boars (Duroc × Saba, DS) and 37 hybrid sows (Yorkshire × (Landrace × Saba), YLS) mated to produce a total of 223 four-way crossbred individuals. All test pigs were raised under identical dietary and environmental settings and had ad libitum access to feed and water. Ear tissue samples were collected from 223 crossbred pigs when they weighed an average of 105.25 ± 15.75 kg.

Ten growth traits, including LBT, ADG30–60, ADG60–100, ADG30–100, BL, BH, BAH, CC, CD, and RC, were measured by following standard procedures. The growth performance measurement of the test pigs started at about 30 kg and ended at about 100 kg. LBT, ADG30–60, ADG60–100, and ADG30–100 were measured according to the method of “Pig Production Performance Determination Regulations” (NY/T822). Collecting three ADG traits phenotypic data was a very laborious and complicated process. Thus, only 198 out of 223 individuals were phenotyped.

All test pigs were measured for body size traits, including BL (from the midpoint of the ears to the base of the tail), BH (from the shoulders to the ground), BAH (from the lowest point of the back to the ground), CC (the circumference around the posterior edge of the scapula), CD (from the withers to the sternum, measures along the posterior edge of the scapula), and RC (from the front edge of the left knee joint to the anus, from the anus to the front edge of the right knee joint) on the flat ground when the weight of the test pigs reached 105.25 ± 15.75 kg. BL, CC and RC were measured with a tape measure, while BH, BAH and CD were measured using a measuring stick.

Proc MEANS in SAS 9.4 (SAS Institute, Inc., Cary, North Carolina) was employed to produce descriptive statistics, including the number, minimum, maximum, mean, standard deviation, and coefficient of variation, of ten growth traits for pig accessions. Using the R package “ggpubr”, the sample distribution was represented as a frequency distribution histogram. The R function “PerformanceAnalytics” was used for the phenotypic correlation analysis. The genetic correlations for three ADG traits were estimated using GCTA software [36].

### 2.3. SLAF Library Construction and High-Throughput Sequencing

Molecular markers throughout the entire genome were generated as described previously [34]. In short, a total of 223 intercross pigs were sequenced using SLAF-seq technology [18]. First, we employed the phenol-chloroform extraction protocol to extract genomic DNA from ear tissue samples. Then, the pig genome (Sscrofa 11.1_102, ftp://ftp.ensembl.org/pub/release-102/ (accessed on 21 November 2021)) as the reference genome was used for an electronic digestion prediction experiment. According to the selection principle of the enzyme digestion scheme [25], the appropriate restriction enzyme combination was determined to digest genomic DNA. After that, a series of operations were performed, including fragment end reparation, paired-end adapter ligation, PCR amplification, purification, and SLAF library construction. Meanwhile, the control genome (*Oryza sativa spp. japonica*; 374.30 Mb; http://rapdb.DNA.affrc.go.jp/ (accessed on 21 November 2021)) was used to verify the reliability of the experimental process. Ultimately, SLAF-seq for each individual was conducted on an Illumina HiSeq 2500 platform (Illumina, Inc., San Diego, CA, USA) at Beijing Biomarker Technologies Corporation in Beijing, China.

### 2.4. Data Processing and SNP Calling

The raw SLAF-seq data were further analyzed using Dual-Index software [37] to obtain the raw paired-end sequencing reads for each accession. Then, we used BWA software [38] to align raw paired-end reads with the pig reference genome (Sscrofa 11.1_102). Afterward, polymorphic SLAF tags were obtained. Based on the polymorphic SLAF tag information, local realignments were conducted, and SNPs were detected using GATK software [39]. SAMtools software [40] was also used to find SNPs, ensuring the accuracy of the SNPs discovered using GATK. The SNPs jointly identified by the two software programs were considered reliable. Ultimately, a total of 227, 921 SNPs were acquired for further study by filtering according to minor allele frequency (MAF: 0.05) and integrity (int: 0.8) using PLINK 2 [41].

### 2.5. Genome-Wide Association Study (GWAS)

A total of 227,921 filtered SNPs detected from 223 accessions were used for GWAS. The mixed linear model (MLM) of GEMMA software [42] was employed for association analysis between growth traits and reliable SNP markers. The MLM formula of GEMMA software was as follows:y = Wα + Xβ + Zμ + ε
where y was the phenotype, X was the genotype, W was the matrix of population structure calculated by the ADMIXTURE software [43], and Z was the matrix of the kinship relationship calculated using GCTA software [44]. α and β were fixed effects, while μ and ε were random effects. Finally, for each variant site, an association result could be attained. The Bonferroni correction approach for multiple testing [42] was particularly strict and could only identify a small number of significant SNPs. Therefore, SNP markers with adjusted *p*-value < 1 × 10^−5^ (−log_10_
*p* > 5, control threshold) were regarded as significant associations with the trait of interest. Based on the number of filtered SNPs (*n* = 227,921), the threshold *p*-value 4.39 × 10^−7^ (0.1/227,921) and 4.39 × 10^−8^ (0.01/227,921) were genome-wide 10% and 1% significance levels, respectively. Finally, the manhattan and Quantile-quantile (Q-Q) plots of GWAS were drawn using the R package “qqman”.

### 2.6. Identification and Annotation of Candidate Genes

Based on the previous references [28,45,46], the genes within 100 kb upstream or downstream of significant associated SNPs were deemed as potential candidate genes for growth traits. The relevant information on these potential genes was downloaded from the Ensembl Sscrofa11.1 database (www.ensembl.org (accessed on 15 December 2021)). Afterward, GO annotation of candidate genes was conducted using Gene Ontology Consortium (http://geneontology.org (accessed on 15 December 2021)).

### 2.7. Association Analysis between SNP Marker Genotypes and Growth Traits

Proc GLM in SAS 9.4 (SAS Institute, Inc., Cary, NC, USA) was used to estimate the associations between SNP marker genotypes and growth traits. Additive genetic effects were calculated by comparing the two homozygous genotypes in pairs, and the dominance effects were computed as the deviation of the heterozygote effect from the mean of the two homozygous genotypes.

## 3. Results

### 3.1. Phenotype Description and Correlation among Growth Traits

The statistical information on the ten growth traits is shown in Appendix A. The mean values for LBT, ADG30–60, ADG60–100, ADG30–100, BL, BH, BAH, CC, CD, and RC were 29.51 mm, 448 g, 556 g, 510 g, 112.23 cm, 63.85 cm, 70.84 cm, 114.81 cm, 36.64 cm, and 115.38 cm, respectively. Coefficients of variation for the ten growth traits were 2.37, 26.58, 22.45, 15.77, 6.94, 5.75, 5.78, 4.68, 7.51, and 8.36, respectively. The results, therefore, indicated that four-way crossbred pig populations had a large variation for three ADG traits. The frequency distributions of the traits are shown in Figure 1. The ten growth traits appear to conform to the normal distribution. The phenotypic correlation coefficients for the ten growth traits are shown in Table 1. The results revealed that BAH had the strongest positive correlation with BH (*r* = 0.70, *p* < 0.001). The genetic correlations for three ADG traits are shown in Table 2. There is a negative genetic correlation between ADG30–60 and ADG60–100 ((*r* = −0.27).

### 3.2. Identification of SLAFs and SNPs

A total of 223 individuals were genotyped and descriptive statistics of the sequence data were presented in our previous study [34]. In short, two restriction enzymes, *RsaI* and *HaeIII*, were chosen as enzyme combinations for the development of SLAF tags in accordance with the selection principle of the enzyme digestion scheme, and the sequence with the length of 314–344 bp was defined as SLAF tags. In total, 1109.92 M paired-end reads were obtained from 223 four-way crossbred pigs. The average value of Q30 (Q30 represented a quality score of 30, indicating an error rate of 0.1% or sequence accuracy of 99.9%) and Guanine-cytosine (GC) content were 90.74% (83.17–94.72%) and 44.83% (41.89–49.18%), respectively, demonstrating that our sequencing results for 223 accessions were reliable (Appendix A). Further analysis revealed that 1552,377 SLAF tags were identified, with 153,084–581,243 SLAFs (average, 331,608) for each accession. The average sequencing depth of all accessions was 11.94 (5.95–25.62 for each accession), which met the assumptions of the SLAF test and guaranteed the accuracy of subsequent analysis (Appendix A). During sequencing, additionally, rice Nipponbare (*Oryza sativa ssp. Japonica*) was employed as a control. The results demonstrated that the construction of SLAF libraries was normal because the enzyme digestion efficiency and paired-end comparison efficiency of control data reached 90.77% and 95.4%, respectively.

In total, 16,997 polymorphic SLAF tags and 10,784,484 SNPs were identified across 233 accessions after genomic mapping and SNP calling. The average value of the number, integrity and heterozygosity ratio of SNPs were 2,216,210 (867,966–4,616,267), 20.55% (8.05–42.8%) and 7.51% (6.66–14.39%), respectively (Appendix A). Furthermore, a total of 227,921 highly consistent SNPs were found after the genotyping results were filtered for a minimum MAF of 0.05 and locus integrity of 0.8. The density distributions of the filtered SNPs across Sscrofa genome are shown in Figure 2. SNPs were found in almost all of the non-overlapping 1 Mb regions of the genome. The density distribution of total SNPs and filtered SNPs were calculated on each Sscrofa autosome and are shown in Appendix A. The filtered SNP density across the 18 Sscrofa chromosomes was one SNP every 10.28 kb on average, which indicated that the data was reliable.

### 3.3. Genome-Wide Association Study and Identification of Candidate Genes

Q-Q plots of all traits were drawn since population stratification could have an impact on GWAS. We found that the observed −log_10_
*p* values of the MLM were fairly close to the expected −log_10_
*p* values. The results showed that the MLM well controlled the research’s false positives. Q-Q plot of each growth trait was shown following the manhattan plot of the corresponding traits (Figure 3, Figure 4 and Figure 5). In total, 53 SNPs were identified as significant (*p* < 1.0 × 10^−5^) for the traits investigated using MLM (Appendix A). The phenotypic variation explained (PVE) by the significant SNPs was from 0.72 to 19.35 (Appendix A). Five, ten, seven, four, five, three, ten, one, five, and three SNPs were significantly associated with LBT, ADG30–60, ADG60–100, ADG30–100, BL, BH, BAH, CC, CD, and RC, respectively. These detected SNPs were distributed in fourteen Sscrofa chromosomes (SSC), except for SSC11, SSC12, SSC15 and SSC16. Moreover, a total of 99 genes located within 100 kb upstream and downstream of these significant SNPs were considered potential candidate genes (Appendix A).

#### 3.3.1. LBT, ADG30–60, ADG60–100, and ADG30–100

GWAS results and candidate genes for LBT, ADG30–60, ADG60–100, and ADG30–100 are showed in Appendix A and Figure 3.

In total, five SNPs distributed on SSC1, SSC4, SSC10, and SSC18 were significantly associated with LBT. On SSC4, two adjacent SNPs (rs322460444 and rs332806988) identified were not located in any genes. SNP rs322460444 exceeded the 10% genome-wide significance level (*p* = 3.53 × 10^−^^7^). The significant SNP on SSC10 was located 29.6 kb upstream of ATP synthase subunit O (mitochondrial, *ATP5O*).

For ADG30–60, two adjacent SNPs (rs334892514 and rs690227348) on SSC18 were located 1.4 and 1.5 kb, respectively, upstream of the growth hormone-releasing hormone receptor (*GHRHR*) gene. On SSC4, three adjacent SNPs (rs325760894, rs81382100 and rs320502793) were associated with ADG30–60. The nearest genes of these three SNPs were corticotropin releasing hormone (*CRH*) and tripartite motif containing 55 (*TRIM55*).

For ADG60–100, two SNPs (rs331585700 and SSC13:94905576) exceeded the 10% genome-wide significance level (*p* = 2.20 × 10^−7^ and *p* = 6.29 × 10^−8^). Among them, the rs331585700 was located 18 kb upstream of eukaryotic translation initiation factor 2 α kinase 1 (*EIF2AK1*). Additionally, the significant SNP on SSC14 related to ADG60–100 was located within pleckstrin homology domain-containing family A member 1 (*PLEKHA1*).

For ADG30–100, one significant SNP (rs1111308563) on SSC14 was located within BRCA1 associated protein (*BRAP*).

#### 3.3.2. BL, BH, and BAH

GWAS results and candidate genes for BL, BH, and BAH are showed in Appendix A and Figure 4.

For BL, all significant SNPs detected were located on SSC7. Among them, the peak SNP (rs335597506) was located within collagen type XI α 2 chain (*COL11A2*). Another significant SNP (rs341689410) was located 27.6 kb and 27.8 kb downstream of nudix hydrolase 3 (*NUDT3*) and high mobility group AT-hook 1 (*HMGA1*), respectively.

For BH, a total of three significant SNPs were identified on SSC6, SSC7, and SSC14. However, these three SNPs were not located in any coding genes.

For BAH, two neighboring SNPs (rs711388225 and rs336803962) on SSC7 were located 6.2 kb downstream of the NHL repeat containing the E3 ubiquitin protein ligase 1 (*NHLRC1*) gene. Another significant SNP (rs318981132) was located 4.4 kb downstream of tripartite motif containing 39 (*TRIM39*). Two adjacent SNPs on SSC14 were located with the small G protein signaling modulator 1 (*SGSM1*) gene.

#### 3.3.3. CC, CD, and RC

GWAS results and candidate genes for CC, CD, and RC are showed in Appendix A and Figure 5.

Only one SNP was detected to be significantly associated with CC. The SNP was located 20.7 kb upstream of nuclear factor of activated T-cells 2 (*NFATC2*).

For CD, two SNPs (rs334022393 and rs705385434) exceeded the 10% genome-wide significance level (*p* = 3.09 × 10^−7^ and *p* = 3.81 × 10^−7^). The most significant SNP (rs334022393) on SSC3 was located 26.5 kb downstream of NCK adaptor protein 2 (*NCK2*), while the significant SNP (rs705385434) on SSC2 was located 2.8 kb upstream of mastermind like transcriptional coactivator 1 (*MAML1*).

For RC, two adjacent SNPs (rs699438879 and rs712077976) on SSC17 were located with the pleckstrin and Sec7 domain containing 3 (*PSD3*).

### 3.4. Comparison with Previously Mapped QTL in Pigs

The Pig Quantitative Trait Locus (QTL) Database (Pig QTLdb, https://www.animalgenome.org/cgi-bin/QTLdb/SS/index, accessed on 25 April 2022) was searched based on SNP and QTL locations to see if QTLs linked to growth traits in this study repeat any previously known QTLs. A total of 53 SNPs significantly associated with growth traits in a four-way crossbred pig population were identified, of which 18 SNPs were located in previously reported QTL regions in pigs. The remaining 35 SNPs had not been included in any previously reported QTLs associated with pig growth traits. Interestingly, two novel QTLs on SSC4 (68.43–70.14 Mb) and SSC7 (25.22–32.32 Mb) were found to be associated with ADG30–60 and BL, respectively. The results of QTLs comparison are shown in Appendix A.

### 3.5. Functional Annotation of Candidate Genes

The results of GO annotation showed that *ATP5O* participated in ATP biosynthetic process. *GHRHR* was involved in positive regulation of multicellular organism growth and positive regulation of growth hormone secretion. *PLEKHA1* participated in post-embryonic development, multicellular organism growth, and skeletal system morphogenesis. The GO annotation results of other genes are shown in Appendix A.

### 3.6. Association Analysis between SNP Marker Genotypes and Growth Traits

The association between candidate SNPs genotypes and growth traits was estimated. The result revealed that genotypes (G–T, C–T, A–G, C–T, and G–T) of five SNPs, including rs325760894, rs81382100, rs320502793, SSC14:42805887, and rs705385434, presented extremely significant associated (*p* < 0.01) with corresponding traits. Genotypes (C–T and A–T) of two SNPs (SSC14:131969638 and SSC14:42805901) presented significant associated (*p* < 0.05) with corresponding traits. The G, T, and G alleles were favorable for bigger ADG30–60 on rs325760894, rs81382100, and rs320502793, respectively. The T allele was favorable for bigger ADG60–100 on SNP SSC14:131969638. The C and A alleles were favorable for higher BAH on SSC14:42805887 and SSC14:42805901, respectively. The T allele was favorable for bigger CD on rs705385434. The effects of the genotypes and additive and dominance effects are shown in Table 3.

## 4. Discussion

### 4.1. QTLs Identified for Growth Traits

A total of 53 SNPs were identified as significant for the growth traits investigated, of which 35 SNPs had not been included in any previously reported QTLs for growth traits of pigs. Furthermore, we identified one novel QTL, which was located in a 7.10-Mb region (25.22–32.32 Mb) on SSC7 significantly associated with BL (Appendix A). In several previous studies, some significant SNPs associated with BL were also found in the 7.10-Mb region in different pig populations [2,13,16,47]. Among all SNPs detected, 18 SNPs were located in previously reported QTL regions for growth traits in pigs. One new QTL detected was located in a 1.71-Mb region (68.43–70.14 Mb) on SCC4 significantly associated with ADG30–60. This region was located in 20 previously reported QTLs related to average daily gain (ADG) (Appendix A), which spanned more than 7.29 Mb. We further narrowed the interval of QTLs for ADG in the current study. Additionally, a 0.32-Mb region (116.90–117.22 Mb) on SSC14 was identified as being significantly associated with BH and BAH, containing the significant SNP SSC14:116903219 for BH, and rs339643700, rs337355885 and rs318249884 for BAH (Appendix A). Another 3.58-Mb region (52.72–56.30 Mb) on SSC17 was identified as being significantly related to ADG60–100 and CC, containing the significant SNP rs694911590 for CC, and rs320122060 for ADG60–100. These results showed that some chromosomal regions might have diverse effects on different growth traits. Moreover, high and low correlation coefficients were found between BH and BAH (*r* = 0.70; *p* < 0.001), and between ADG60–100 and CC (*r* = 0.28; *p* < 0.001) (Table 1), respectively. The correlation coefficients provided an indication of pleiotropic effects.

### 4.2. Candidate Genes for LBT, ADG30–60, ADG60–100 and ADG30–100

The study showed that the significant SNP on SSC10 related to LBT was located 29.6 kb upstream of *ATP5O*. The results of GO annotation revealed that *ATP5O* participates in ATP biosynthetic process. A study found that *ATP5O* played a role in the regulation of glucose metabolism in vivo [48].

Two adjacent SNPs on SSC18 associated with ADG30–60 were located 1.4 and 1.5 kb, respectively, upstream of the *GHRHR* gene. GO annotation results indicated that *GHRHR* is mainly involved in positive regulation of multicellular organism growth and positive regulation of growth hormone secretion. The biological process is that growth hormone-releasing hormone (GHRH) stimulates pituitary growth hormone (GH) synthesis and secretion as well as somatotroph proliferation in mammals, which is mediated by *GHRHR* [49,50,51,52]. Two studies found that GH mediates the skeletal muscle development of mice [53] and stimulates skeletal muscle growth in cattle [54]. Previous studies showed that treatment with growth hormone (GH) affected pig growth performance, resulting in an increase in the average daily gain [55,56,57]. Similarly, mutations in *GHRH* and *GHRHR* in other mammals also influenced growth and development [58]. Armstrong et al. reported that SNP RS400358099 in *GHRHR* regulates the growth traits of Texel lambs [59]. Thus, *GHRHR* regulated skeletal muscle growth and development by mediating synthesis and secretion of GH and should be considered a strong candidate gene for porcine ADG30–60.

Furthermore, the GWAS result showed that three significant SNPs on SSC4 were associated with ADG30–60. Genotypes (G–T, C–T and A–G) of these three SNPs presented extremely significant association (*p* < 0.01) with ADG30–60. The ADG30–60 of GG, TT and GG genotypes were 52~80 g more than GT, CT and AG genotypes, and 139~152 g more than TT, CC and AA genotypes (Table 3). The nearest genes of the three SNPs were *CRH* and *TRIM55*. In its role as a crucial stress response regulator, CRH generates a series of biological effects through its receptors, mobilizes various body systems to respond to stress stimuli, and controls endocrine, immunological, and behavioral responses [60]. *TRIM55* is a member of the TRIMs family. The TRIM motif is made up of zinc-binding domains, a B-box motif, a RING finger region, and a coiled-coil domain. Among the TRIMs family, *TRIM55* is also a known muscle-specific RING finger (MURF). This unique subclass of RING Finger-B box-coiled-coiled (RBCC)/tripartite motif (TRIM) proteins contain two coiled-coil dimerization motif boxes, a zinc-binding B-box motif, and a highly conserved N-terminal RING domain [61]. The RING-finger involved in protein-protein interactions is an unusual type of zinc-binding Cys-His protein motif, which is found in an increasing number of proteins and plays a role in signal transduction, gene transcription, ubiquitination, morphogenesis, and differentiation [62,63]. By interacting with myofibril components (including the giant protein and titin), microtubules, and/or nuclear factors, TRIM55 proteins function as cytoskeletal adaptors and signaling molecules [64]. McElhinny et al. [65] demonstrated that the knockdown of antisense oligonucleotides of *TRIM55* resulted in delayed myoblast fusion and myofibrillogenesis and affected contractile activity. Zhang et al. [66] suggested that *TRIM55* is a promising candidate gene for traits related to skeletal muscle development since it is developmentally regulated and likely has a significant function during the embryonic and early postnatal skeletal development phases. Given that *TRIM55* is functionally related to skeletal muscle development, it might be a leading candidate gene for the locus.

For ADG60–100, the significant SNP (rs331585700) was located 18 kb upstream of the *EIF2AK1* gene, which plays an important role in regulating protein synthesis in response to stress. GO annotation results showed that *EIF2AK1* participates in response to external stimulus and negative regulation of cell proliferation (Appendix A). Gong et al. found that *EIF2AK1* was associated with growth traits of Chinese Bamaxiang pigs and was considered a plausible candidate gene for body mass index (BMI) [13]. In addition, the significant SNP (SSC14:131969638) was located within *PLEKHA1*. Genotypes (C–T) of the SNP presented significant association (*p* < 0.05) with ADG60–100. The ADG60–100 of TT Genotype was 226 g and 134 g more than of CC and CT, respectively (Table 3). GO annotation results showed that *PLEKHA1* participates in post-embryonic development, multicellular organism growth, and skeletal system morphogenesis. The gene *PLEKHA1* also known as *TAPP1*, encodes an adaptar protein containing the pleckstrin homologous domain. The protein was crucial for remodeling the actin cytoskeleton in response to growth factor stimulation [67]. Kim et al. found that *PLEKHA1* might be related to the body size of Yucatan miniature pigs [68]. Additionally, the *PLEKHA1* gene was prominently related to the PI3K/Akt signaling pathway, which was crucial for osteoporosis [69,70]. By controlling PtdIns (3,4,5) P3, the *PLEKHA1* gene might be implicated in bone metabolism [71]. Collectively, the *PLEKHA1* gene should be regarded as a strong candidate gene for ADG60–100.

For ADG30–100, the significant SNP (rs1111308563) was located with the *BRAP* gene. GO annotation results showed that *BRAP* participates in MAPK cascade and ras protein signal transduction (Appendix A). It is well known that Ras is a well-known key upstream regulator of the MAPK. The Ras-MAPK signaling pathway can control cell proliferation, differentiation, and survival through the kinase cascade [72,73,74]. Many studies have shown that Ras-MAPK signaling pathway is crucial for adipogenesis [75,76]. Therefore, it could be inferred that *BRAP* gene might regulate fat deposition in pigs through Ras-MAPK signal pathway, thereby affecting average daily gain of pigs.

### 4.3. Candidate Genes for BL and BAH

For BL, the peak SNP (rs335597506) on SSC7 was located within *COL11A2*. Zhou et al. [77] found that *COL11A2* was significantly associated with body shape of Large Yellow Croaker. Some research showed that the *COL11A2* gene was related to the skeletal system morphogenesis and growth of cartilage [78,79]. Another study revealed that the *COL11A2* gene responsible for skeletal system morphogenesis and body growth was differentially expressed in adipose tissue between Jeju native pigs and Berkshire [80]. Moreover, the size of newborn homozygote *COL11A2* mutant mice is 25% lower than that of wild-type counterparts [81]. Gao et al. [82] found that *COL11A2* was related to the body height of the Chongming white goat. Therefore, the *COL11A2* gene should be regarded as a strong candidate gene for BL. Furthermore, another significant SNP (rs341689410) on SSC7 for BL was located 27.6 kb and 27.8 kb downstream of *NUDT3* and *HMGA1*, respectively. The *NUDT3* gene participates in the catabolism of diadenosine polyphosphate, and variations in this gene have been linked to human BMI [83] and human height [84]. The *HMGA1* gene is one member of the high mobility group A family. The fact that *HMGA1*/*HMGA2* double knock-out mice are smaller than *HMGA1* knock-out mice suggests that *HMGA1* can influence the body size of animals [85]. A study showed that *HMGA1* could operate as a glucose disposal mediator by controlling the activity of insulin-like growth factor 1 [86]. It was also significantly associated with human height [87]. Furthermore, numerous studies have demonstrated a connection between *HMGA1* and pig body size traits [47]. Through GWAS, some research discovered that *HMGA1* was a candidate gene for several body size traits of Yorkshire pigs [3], White Duroc × Erhualian F2 intercross pigs [16], and Chinese Bamaxiang pigs [13]. According to Zhang et al. [88], *HMGA1* was expressed in pig limb cells and had an impact on chondrocyte proliferation and differentiation, so *HMGA1* was considered a candidate gene for limb bone length in a Large White × Minzhu intercross population. Given the functional significance of *HMGA1* and the numerous studies that have demonstrated how strongly it was correlated with body size features, *HMGA1* should be regarded as a strong candidate gene for pig body size traits, including BL.

For BAH, two neighboring SNPs on SSC7 were located 6.2 kb downstream of the *NHLRC1* gene. This gene encodes the malin protein containing a zinc-binding RING finger motif with E3-ubiquitin ligase activity [89,90], which is involved in regulating the biosynthesis of glycogen [91]. Two studies found that the *NHLRC1* gene was related to the body height of Sahiwal cattle [92] and Chinese Holstein cattle [93]. Another significant SNP (rs318981132) was located 4.4 kb downstream of *TRIM39*, which is also a member of the RBCC/TRIM subfamily of zinc finger proteins that participated in a variety of biological processes, including cell differentiation [94]. One study found that a SNP (AX_101003762) of the *TRIM39* gene was significantly associated with body weight at 28 days of age in broilers [95]. In addition, two adjacent SNPs (SSC14:42805887 and SSC14:42805901) were related to BAH. Genotypes (C–T and A–T) of these two SNPs presented extremely significant (*p* < 0.01) and significant association (*p* < 0.05) with BAH, respectively. Among them, the BAH of CC genotype of SSC14:42805887 was 2.21 cm and 5.75 cm higher than that of CT and TT, respectively, and the BAH of AA genotype of SSC14:42805901 was 1.99 cm and 6.73 cm higher than that of AT and TT, respectively. The two adjacent SNPs were located within the *SGSM1* gene, which participates in regulating cell circle, proliferation and differentiation [96]. According to a study, a SNP (rs336761069) of the *SGSM1* gene was significantly related to the chest circumference of Chinese Sushan pigs [97].

### 4.4. Candidate Genes for CC, CD and RC

Only one SNP was significantly associated with CC. The location of the SNP was 20.7 kb upstream of *NFATC2*. It was discovered that the *NFATC2* gene contributed to the growth development of skeletal muscle via the PGF2 receptor [98]. Myoblasts must fuse to produce multinucleated myofibers or myotubes for the growth and development of skeletal muscle. After a myotube had initially formed, *NFATC2* regulated myoblast fusion, which was extremely important for further cell growth [99]. A study found that an allele of *NFATC2* was significantly associated with birth weight and body weight at the 8th week of age (weaning weight) of crossbred pigs [100]. Thus, the *NFATC2* gene could be deemed as a potential candidate gene for CC.

For CD, the most significant SNP was located 26.5 kb downstream of the *NCK2* gene. The previous work identified *NCK2* as a new regulator of adiposity and suggested that *NCK2* was crucial for preventing white adipose tissue expansion and dysfunction in mice and humans [101]. In Angus cattle, the *NCK2* gene has been located in four QTLs for fat thickness, marbling score, yearling and mature body weight [102]. Moreover, a significant SNP (rs705385434) on SSC2 was associated with CD. Genotypes (G–T) of the SNP presented extremely significant association (*p* < 0.01) with CD. The CD of GG genotype was 2.23 cm and 6.94 cm less than that of GT and TT, respectively. The SNP was located 2.8 kb upstream of *MAML1*. GO annotation results showed that *MAML1* participates in positive regulation of myotube differentiation and myoblast differentiation (Appendix A). A study revealed that mice with a *MAML1* targeted disruption exhibited severe muscular dystrophy. In vitro, MyoD-induced myogenic differentiation did not occur in *MAML1*-null embryonic fibroblasts. The study demonstrated that *MAML1* functioned as a coactivator for *MEF2C* transcription and was required for normal muscle growth [103]. It was inferred that *MAML1* could impact CD by affecting muscle development and should be considered a strong candidate gene for CD.

Finally, for RC, two adjacent SNPs on SSC17 were located with *PSD3*. The *PSD3* gene encodes a protein with a Sec7 domain and a pleckstrin domain. The Sec7 domain is a guanine nucleotide exchange factor, which is an essential element of intracellular signaling networks, while the pleckstrin domain, which may bind phosphatidylinositol, G proteins, and protein kinase C, is present in a wide variety of proteins and functions as a scaffold protein in signal transduction pathways. *PSD3* is one of the 8p22 linkage region potential genes for BMI [104] and childhood and adolescent obesity [105]. Gong et al. [106] found that common variants within the *PSD3* gene were associated with obesity, type 2 diabetes, and high-density lipoprotein cholesterol level. Perhaps, the *PSD3* gene might be used as a potential candidate gene for RC.

However, in further research, more pig populations must be used to confirm these loci and genes, and additional pig biological experiments must be conducted to confirm their roles and functions.

### 4.5. Comparison between ADG30–60 and ADG60–100

In the study, ADG in three stages (30–60 kg, 60–100 kg and 30–100 kg) were measured. Genetic correlations between ADG30–60 and ADG30–100, and between ADG60–100 and ADG30–100 were more than 0.87, while the genetic correlation between ADG30–60 and ADG60–100 was −0.27 (Table 2), which indicated the two traits had weak negative genetic correlation. As we know, pigs at 30–60 kg and 60–100 kg stages have different developmental characteristics. Pigs in the early growth period (30–60 kg) are mainly characterized by the growth and development of bone and muscle, while pigs in the late growth period (60–100 kg) are mainly characterized by fat deposition [107,108,109,110]. Etherton et al. found that pigs began to accumulate a lot of body fat from 45 kg weight. Between 45 kg and 110 kg weight, the fat content increased 10 times disproportionately [111]. At present, the pathways that link regulation of muscle and fat formation are not well understood. Research for new genetically engineered mice showed that an inverse relationship existed between the control of myogenesis/hypertrophy and adipogenesis [112]. As we known, the muscle growth ability of Western lean-type pig breeds was stronger than that of Chinese native fat-type pig breeds, but the fat deposition ability was weaker than that of Chinese fat-type pigs.

In the present study, the average of ADG30–60 (448 g) was 108 g less than ADG60–100 (556 g), indicating that the average daily gain of pigs was more in the 60–100 kg stage. Thus, the rapid fat deposition of pigs in the 60–100 kg stage might lead to bigger daily gain. Through gene mapping and functional annotation, *GHRHR* and *TRIM55* detected were candidate genes for ADG30–60, which were closely related to the growth and development of skeletal muscle. However, *EIF2AK1* and *PLEKHA1* detected were candidate genes for ADG60–100, which were associated with body mass index and post-embryonic development and skeletal system morphogenesis, respectively. Furthermore, GWAS results showed that no identical significant SNP was identified for the two traits. Ten SNPs distributed in SSC4, SSC8, and SSC18 were significantly associated with ADG30–60, while seven SNPs distributed in SSC1, SSC2, SSC3, SSC13, SSC14, and SSC17 were significantly related to ADG30–60. Collectively, the growth of pigs in the two periods might be regulated by different genomic regions and different functional genes.

## 5. Conclusions

In conclusion, we conducted a GWAS based on SLAF-seq for ten growth traits in 223 four-way crossbred pigs using MLM. A total of 53 significant SNPs, two novel QTLs on SSC4 and SSC7, and 13 candidate genes (*ATP5O*, *GHRHR*, *TRIM55*, *EIF2AK1*, *PLEKHA1*, *BRAP*, *COL11A2*, *HMGA1*, *NHLRC1*, *SGSM1*, *NFATC2*, *MAML1*, and *PSD3*) were identified as being associated with growth traits of pigs. The growth of pigs in the 30–60 kg and 60–100 kg stages might be regulated by different genomic regions and different functional genes. Overall, our study provided new evidence that multiple genes were involved in regulating growth traits in pigs. These SNPs and corresponding candidate genes served as a biological foundation for improving growth and production performance in swine breeding.

## Figures and Tables

**Figure 1 genes-13-01990-f001:**
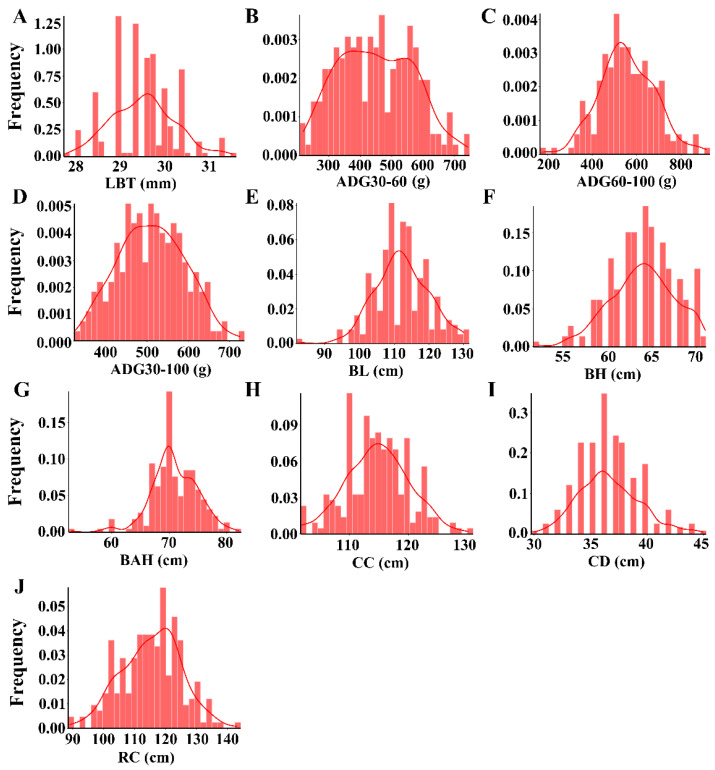
Frequency distribution histogram for ten growth traits. (**A**) Live backfat thickness (LBT), (**B**) Average daily gain from 30 to 60 kg (ADG30–60), (**C**) Average daily gain from 60 to 100 kg (ADG60–100), (**D**) Average daily gain from 30 to 100 kg (ADG30–100), (**E**) Body length (BL), (**F**) Body height (BH), (**G**) Back height (BAH), (**H**) Chest circumference (CC), (**I**) Chest depth (CD), (**J**) Rump circumference (RC).

**Figure 2 genes-13-01990-f002:**
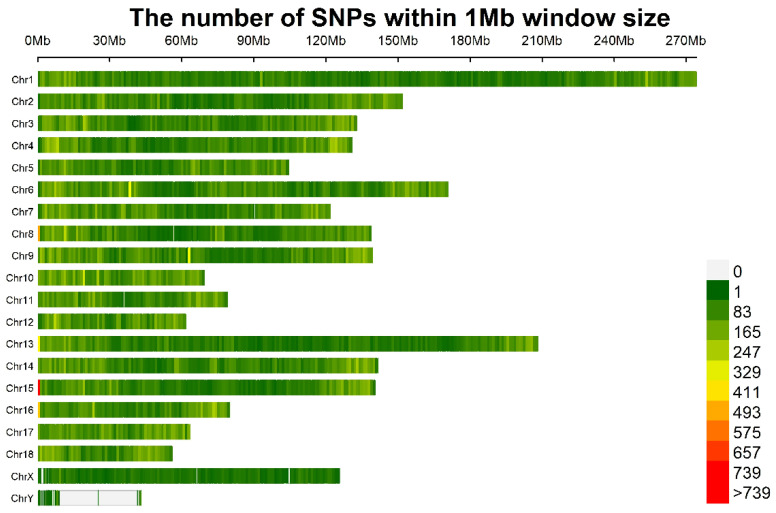
The filtered SNP density distributions on Sscrofa chromosomes. The horizontal axis (*x*-axis) shows the chromosome length (Mb). Color index indicates the number of labels.

**Figure 3 genes-13-01990-f003:**
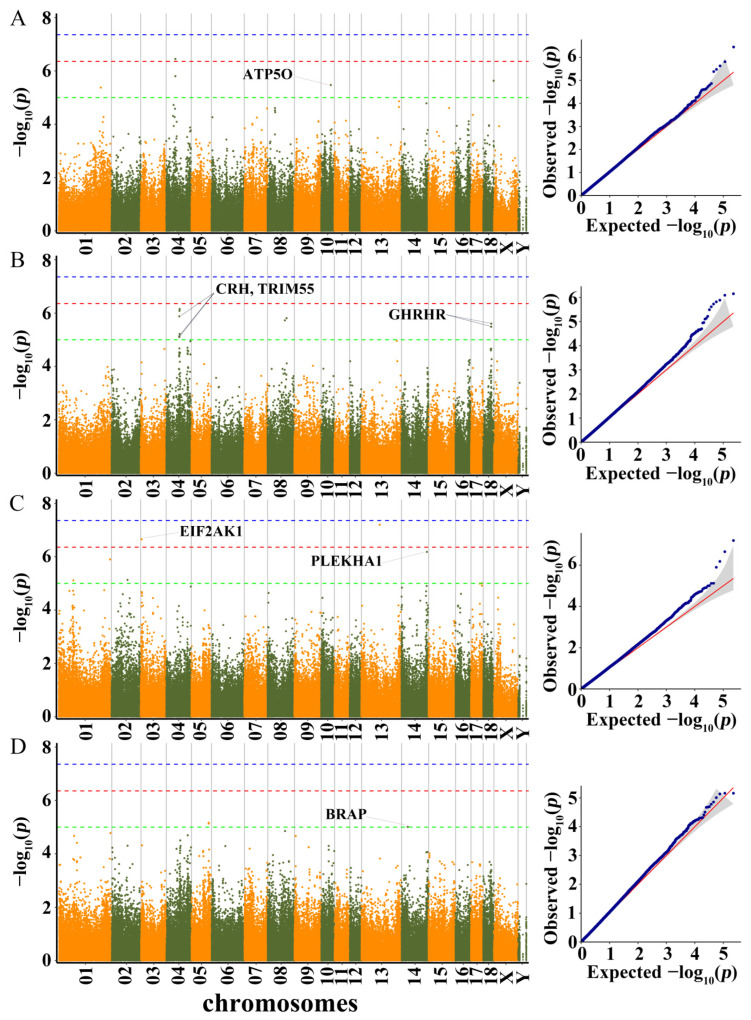
Manhattan plots and QQ plots for LBT, ADG30–60, ADG60–100, and ADG30–100 using MLM. (**A**) LBT, (**B**) ADG30–60, (**C**) ADG60–100, (**D**) ADG30–100. Negative log_10_
*p*-values of the filtered high-quality SNPs were plotted against their genomic positions. The dashed lines of green, orange and blue correspond to the Bonferroni-corrected thresholds of *p* = 1.00 × 10^−5^ (−log_10_ *p* = 5), *p* = 4.39 × 10^−7^ (−log_10_ *p* = 6.36) and *p* = 4.39 × 10^−8^ (−log_10_ *p* = 7.36), respectively.

**Figure 4 genes-13-01990-f004:**
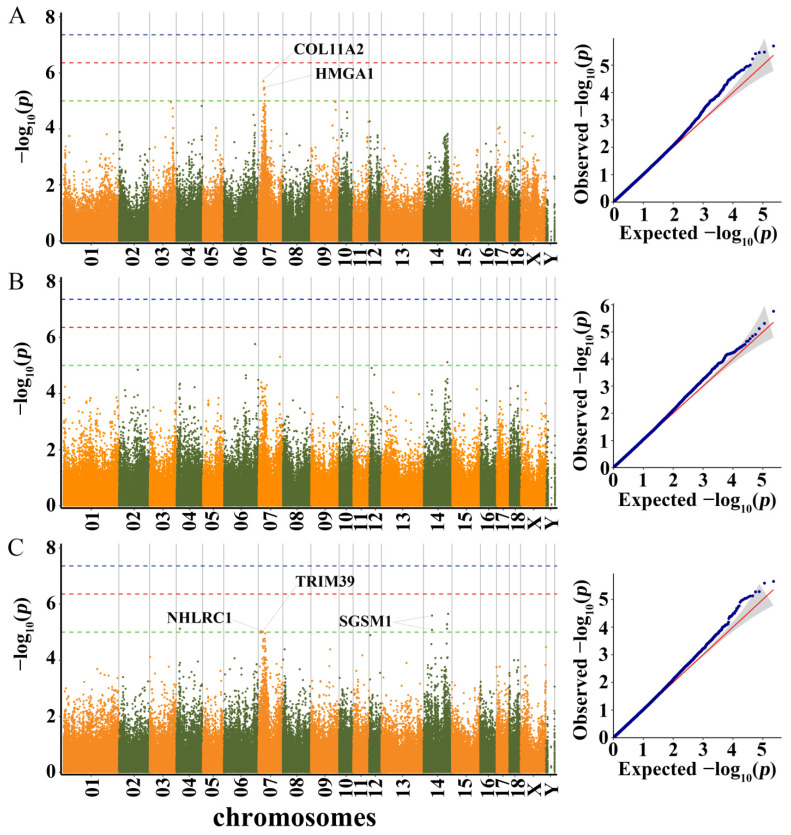
Manhattan plots and QQ plots for BL, BH and BAH using MLM. (**A**) BL, (**B**) BH, (**C**) BAH. Negative log_10_ *p*-values of the filtered high-quality SNPs were plotted against their genomic positions. The dashed lines of green, orange and blue correspond to the Bonferroni-corrected thresholds of *p* = 1.00 × 10^−5^ (−log_10_ *p* = 5), *p* = 4.39 × 10^−7^ (−log_10_ *p* = 6.36) and *p* = 4.39 × 10^−8^ (−log_10_ *p* = 7.36), respectively.

**Figure 5 genes-13-01990-f005:**
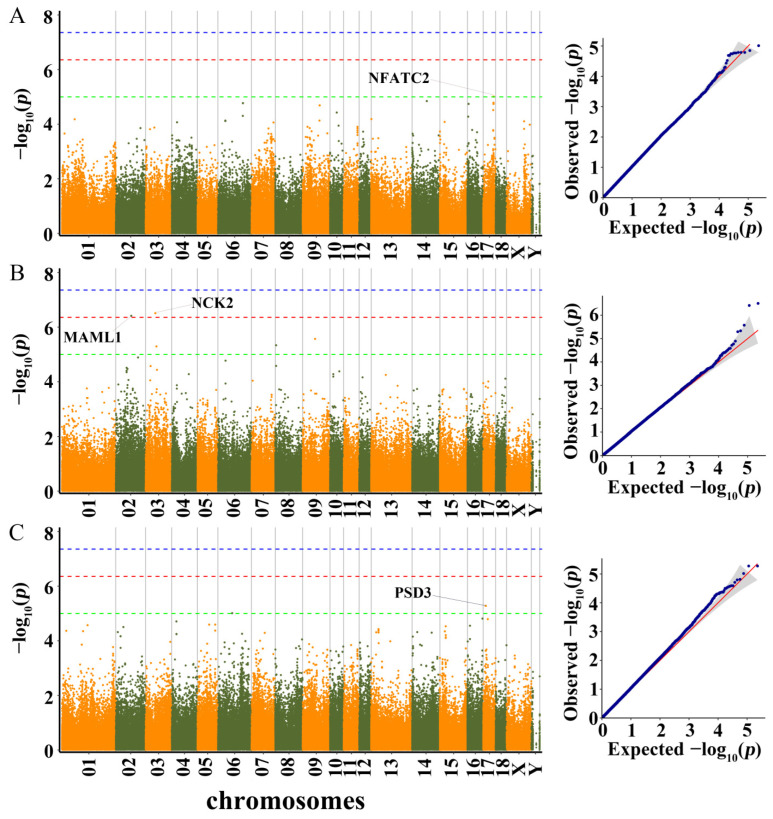
Manhattan plots and QQ plots for CC, CD and RC using MLM. (**A**) CC, (**B**) CD, (**C**) RC. Negative log_10_
*p*-values of the filtered high-quality SNPs were plotted against their genomic positions. The dashed lines of green, orange and blue correspond to the Bonferroni-corrected thresholds of *p* = 1.00 × 10^−5^ (−log_10_ *p* = 5), *p* = 4.39 × 10^−7^ (−log_10_ *p* = 6.36) and *p* = 4.39 × 10^−8^ (−log_10_ *p* = 7.36), respectively.

**Table 1 genes-13-01990-t001:** Phenotypic correlation coefficients among ten growth traits in crossbred pigs.

Trait *	LBT	ADG30–60	ADG60–100	ADG30–100	BL	BH	BAH	CC	CD
ADG30–60	0.29 ***								
ADG60–100	0.19 *	−0.03							
ADG30–100	0.29 ***	0.58 ***	**0.76** ***						
BL	0.01	0.18 *	0.25 ***	0.34 ***					
BH	−0.07	0.04	0.17 *	0.20 **	0.41 ***				
BAH	0.06	0.18 *	0.23 **	0.36 ***	0.54 ***	0.70 ***			
CC	0.10	0.04	0.28 ***	0.25 ***	0.26 ***	0.21 **	0.17 **		
CD	−0.12 *	−0.15 *	0.10	−0.02	−0.06	0.33 ***	0.20 **	0.12 *	
RC	0.01	0.05	0.16 *	0.24 ***	0.46 ***	0.21 **	0.37 ***	0.34 ***	−0.12 *

* LBT: Live backfat thickness, ADG30–60: Average daily gain from 30 to 60 kg, ADG60–100: Average daily gain from 60 to100 kg, ADG30–100: Average daily gain from 30 to100 kg, BL: Body length, BH: Body height, BAH: Back height, CC: Chest circumference, CD: Chest depth, RC: Rump circumference; * significant at *p* < 0.05, ** significant at *p* < 0.01, *** significant at *p* < 0.001; The maximum value of the correlation coefficient between traits is in bold.

**Table 2 genes-13-01990-t002:** Genetic correlations among three ADG traits in crossbred pigs.

Trait Pair *	Genetic Correlation	Standard Error
ADG30–60/ADG60–100	−0.27	0.13
ADG30–60/ADG30–100	0.87	0.10
ADG60–100/ADG30–100	0.93	0.05

* ADG30–60: Average daily gain from 30 to 60 kg, ADG60–100: Average daily gain from 60 to100 kg, ADG30–100: Average daily gain from 60 to100 kg.

**Table 3 genes-13-01990-t003:** Effect of the genotypes on growth traits.

Trait ^a^	SNP ^b^	SSC ^c^	Position (bp) ^d^	Genotype ^e^	N ^f^	Value ^g^	Additive Effect	Dominance Effect
ADG30–60 (g)	rs325760894	4	68428526	GG	34	536.39 ± 18.59 ^A^	75.63	−4.36
				GT	101	456.43 ± 10.79 ^B^		
				TT	57	385.13 ± 14.36 ^C^		
				NN	6			
	rs81382100	4	68463657	CC	67	384.42 ± 13.24 ^C^	67.43	14.90
				CT	85	466.75 ± 11.76 ^B^		
				TT	42	519.28 ± 16.72 ^A^		
				NN	4			
	rs320502793	4	68493463	AA	58	382.11 ± 14.14 ^C^	69.53	0.70
				AG	84	452.34 ± 11.75 ^B^		
				GG	52	521.17 ± 14.93 ^A^		
				NN	4			
ADG60–100 (g)		14	131969638	CC	146	545.88 ± 9.88 ^cB^	88.24	−21.13
				CT	19	637.99 ± 27.40 ^bA^		
				TT	4	772.36 ± 59.71 ^aA^		
				NN	29			
BAH (cm)		14	42805887	CC	138	71.44 ± 0.33 ^A^	2.88	0.67
				CT	35	69.23 ± 0.65 ^B^		
				TT	13	65.69 ± 1.07 ^C^		
				NN	37			
		14	42805901	AA	153	71.16 ± 0.32 ^aA^	3.37	1.38
				AT	24	69.17 ± 0.81 ^bA^		
				TT	7	64.43 ± 1.50 ^cB^		
				NN	39			
CD (cm)	rs705385434	2	78815567	GG	180	36.06 ± 0.18 ^A^	3.47	−1.24
				GT	21	38.29 ± 0.53 ^B^		
				TT	3	43.00 ± 1.40 ^C^		
				NN	19			

^a^ ADG30–60: Average daily gain from 30 to 60 kg, ADG60–100: Average daily gain from 60 to 100 kg, BAH: Back height, CD: Chest depth; ^b^ SNP: rs ID from Ensembl; ^c^ SSC: Sscrofa chromosome; ^d^ Positions of the significant SNP according to the Sscrofa Build 11.1 assembly; ^e^ NN represents no genotype; ^f^ N: Number of pig accessions; ^g^ Different capital letters indicate an extremely significant difference (*p* < 0.01), and different lowercase letters indicate a significant difference (*p* < 0.05).

## Data Availability

The genome sequencing raw data was deposited in NCBI’s SRA database https://trace.ncbi.nlm.nih.gov/Traces/sra/sra.cgi?view=studies&f=study&term=&go=Go; Accession: SRP376933).

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
