# Peer review of "Genome-Wide Association Study of Growth Traits in a Four-Way Crossbred Pig Population"

_genes, 2022, doi:10.3390/genes13111990_

Round 1

Reviewer 1 Report

The authors carried out a GWAS based on SLAF-seq for 9 growth traits in 4-way cross pigs. Forty-nine SNPs were significantly associated with the traits included in the study.  Two novel QTLs and 12 candidate genes were identified.  The procedures utilized appeared to be appropriate to accomplish the study goals.  This manuscript will make a useful contribution to the literature concerning the genetic regulation of growth in swine.  Many changes in wording are needed to improve the grammar and readability of the paper.

lines 15-16: Change to 'Growth traits are crucial economic traits in the commercial pig industry and have a substantial impact on pig production'.

line 23:  Change to '...SSC7 were related to average daily gain from 30 to 60 kg...'.

line 26:  I recommend deleting the word 'dramatically'.  It is rather subjective whether comprehension will be 'dramatically' increased.

line 37:  Change to '...swine industry and have a...'.

line 38:  Change to '...are vital indicators of the growth rate of pigs...'.

lines 42-44:  Change to ‘Both genetic and non-genetic effects, including pig breed, feeding behavior, and nutrition level, influence growth traits of pigs’.

line 44:  Change to ‘…pig growth performance has been…’.

lines 46-47:  Change to ‘…numerous minor genes.  Therefore, improving growth traits through conventional breeding is time-consuming and costly.

lines 47-49:  Change to ‘Due to the rapid development of molecular markers and the completion of the pig genome sequence, molecular breeding has become an effective way to improve growth traits’.

lines 50-51:  Change to ‘…total of 2,597 quantitative trait loci (QTL) associated with growth traits have been added to the pig QTL database…’.

line 60: ‘has’ in place of ‘had’.

line 62:  Delete ‘in pigs’.  It is clear that you are talking about pigs.

line 67:  Delete ‘besides’.

lines 71-72:  Change to ‘…sequencing (WGS) is another genotyping method that has been used in recent years…’.

line 76:  Delete ‘extremely’.

line 79:  Change to ‘Abundant SNPs and candidate genes have been identified…’.

line 82:  Delete ‘successfully’.

line 85:  Delete ‘pig’.

line 88:  Delete ‘As we know’.

line 89:  Change to ‘…are significantly different from Chinese native pigs…’.

line 98:  Change ‘during’ to ‘from’ in both places.

line 103:  Change to ‘…this is the first report…’.

line 104: ‘offer’ in place of ‘offered’.

line 117:  Change to ‘…and had ad libitum access to feed and water’.

line 118: Change to ‘…when they weighed an average of 105.25 ± 15.75 kg’.

line 119: Delete ‘In the present study,’.  It is clear that you are talking about the present study.

line 123:  Delete ‘Besides’.

line 126:  Delete ‘the length of’.

line 129: ‘through the anus’?

line 129:  Delete ‘same’.

line 130:  Delete ‘among them’.

line 131: Change ‘sample’ to ‘pig”.

lines 134-135:  What was the statistical model used to analyze the dependent variables?

line 137:  Change to ‘…was used for the phenotypic…’.

line 140: ‘employed’ in place of ‘employ’.

lines 141, 154, and 182: Delete ‘And’.  Do not begin sentences with ‘And’.

line 157: ‘were’ in place of ‘was’.

lines 169-170: I believe it would be more correct to state, ‘α and β were fixed effects, while μ and ε were random effects’.

line 174: Change to ‘…with the trait of interest’.

lines 174-175:  Change to ‘Based on the number of filtered SNPs (n = 227,921), the threshold…’.

line 176: Change to ‘…were the genome-wide 10% and 1% significance levels, respectively’.

line 179: ‘references’ in place of ‘reference’.  You have listed more than one reference.

line 187:  Change to ‘is shown’.

line 190:  Change to ‘coefficients of variation’.

line 191: Change to ‘…7.51, and 8.36%, respectively…’.  The CV is expressed as a percent.

Table S1:  Change the last column heading to ‘CV, %’.

lines 192-193: Change to ‘…crossbred pig populations had a large variation for ADG’.  The CV of 26.58% is large, but I would not call it ‘extraordinary’.  The CVs are an indication of phenotypic variation, not genetic variation.

lines 193-194:  Change to ‘The frequency distributions of the traits are shown in Figure 1. The nine growth traits appear to conform to the normal…’.

line 195: The phenotypic correlation coefficients for the nine growth traits are shown in Table 1’.

Table 1 and in the text: It is not necessary to report the correlation coefficients to 4 decimal places.  Two decimal places are sufficient.

line 197:  Delete ‘Besides’.

line 201: Change ‘during’ to ‘from’ in both places.

line 204:  Change the title of Table 1 to ‘Phenotypic correlations among nine growth traits in crossbred pigs’.

lines 205-206: Change ‘during’ to ‘from’ in both places.

Tables S2 and S3:  Change ‘each accessions’ to ‘each accession’ in the title.

line 223: ‘normally efficiency’ does not make sense.  Please check it.

line 227:  What is ‘heter ratio’?

line 229: Change to ‘…were filtered…’.

line 233: It does not make sense to say, ‘which indicated that the data was available’.  Please reword.

lines 240-241:  Which column of Table S6 shows the percentage of phenotypic variation explained?

line 241: ‘Among them’?  Among what?

line 251: Change to ‘…in any genes.  SNP rs322460444 exceeded…’.

lines 252-253:  Change to ‘…was located 29.6 kb upstream of ATP synthase…’.

line 254: Change to ‘…on SSC18 were located 1.4 and 1.5 kb, respectively, upstream of the growth hormone-releasing hormone receptor (GHRHR) gene’.

line 260: Change to ‘…was located 18 kb upstream of…’.

line 271:  Change to ‘were located’.

line 273:  Change to ‘…was located 27.6 kb and 27.8 kb downstream of nudix…’.

line 277: Change to ‘…on SSC7 were located 6.2 kb downstream of the NHL…’.

line 279:  Change to ‘…was located 4.4 kb downstream of tripartite motif…’.

line 280: Change to ‘Two adjacent SNPs on SSC14 were located within the small G protein…’.

line 289: Delete ‘In this study’.  It is clear that you are presenting results from this study.

line 290: Change to ‘…was located 20.7 kb upstream of the nuclear factor…’.

line 291: Delete ‘identified’.

lines 293-297: Change to ‘…was located 26.5 kb downstream of NCK adaptor protein 2 (NCK2), while the significant SNP (rs705385434) on SSC2 was located 2.8 kb upstream of mastermind like transcriptional coactivator 1 (MAML1). For RC, two adjacent SNPs (rs699438879 and rs712077976) on SSC17 were located within the pleckstrin and Sec7 domain containing the 3…’.

line 308: Delete ‘The study revealed that’.

line 314: Change to ‘are shown’.

Table S7: Delete ‘and newly significant SNPs for growth traits’ from the title of the table.

line 320: Change to ‘are shown’.

line 323: Delete ‘In this study’.  It is clear that you are discussing the results of your study.

lines 331-332: Avoid using two phrases beginning with ‘which’ in the same sentence.

line 336: Delete ‘Besides’.

lines 342-343: The correlation coefficients provide an indication of pleiotropic effects, but they do not really ‘explain’ the pleiotropic effects.

lines 345-346: Change to ‘…was located 29.6 kb upstream …’.

line 348: Delete ‘Besides’.

line 349: Change to ‘…were located 1.4 and 1.5 kb, respectively, upstream of the GHRHR gene…’.

lines 350 and 354: Change ‘was’ to ‘is’.

line 353: Change to ‘…biological process is that growth…’.

lines 355-356: ‘sensitively’?  Do you mean ‘sensitivity’?

lines 357-358: Delete ‘of animals’.

line 358: Change to ‘…GHRHR regulates…’.

line 375:  Change ‘functioned’ to ‘function’ as they still function in this fashion. If you say ‘functioned’, it implies that they used to function in this way in the past, but they no longer do so.

lines 380-383: Change to ‘…since it is developmentally regulated and likely has a significant function during the embryonic and early postnatal skeletal development phases. Given that TRIM55 is
functionally related to …’.

line 384:  Change to ‘…was located 18 kb upstream of the…’.

line 385: ‘plays’ in place of ‘played’.  Use the present tense.

line 386: ‘participates’ in place of ‘participated’.  Use the present tense in place of the past tense since it still plays this role.  These types of changes need to be made throughout the entire manuscript.

lines 390 and 458:  Change to ‘GO’.

line 412:  Change to ‘…BL is located 27.6 kb and 27.8 kb downstream of NUDT3 and HMGA1, respectively’.

line 416: Delete ‘only’.

line 417: Delete ‘besides’.

line 422: ‘Pigs’ does not need to be capitalized.

line 430: Change to ‘SNPs on SSC7 were located 8.2 kb downstream of…’.

lines 431-432: Do not use phrases beginning with ‘which’ twice in the same sentence.

line 435: Change to ‘…located 4.4 kb downstream of…’.

line 437: Replace ‘one research’ with ‘one study’.

line 439: Change to ‘…28 days of age in broilers…’.

lines 444-445: Change to ‘Only one SNP was significantly associated with CC. The location of the SNP was 20.7 kb upstream of NFATC2’.

lines 446-447: Delete ‘As everyone knows’.

 line 453: Change to ‘…the most significant SNP was located 26.5 kb downstream of the NCK2 gene’.

line 458: Change to ‘…on SSC2 was located 2.8 kb upstream of MAML1…’.

lines 484-488: Change to ‘Overall, our study provided new evidence that multiple genes are involved in regulating growth traits in pigs. The SNPs and corresponding candidate genes serve as a biological foundation for improving growth and production in swine breeding’.

line 506: Change to ‘Every effort was made…’.

Author Response

Dear reviewer:

We are very grateful for your objective scientific comments. We have carefully revised the manuscript point by point as advised.

Comment 1: The authors carried out a GWAS based on SLAF-seq for 9 growth traits in 4-way cross pigs. Forty-nine SNPs were significantly associated with the traits included in the study.  Two novel QTLs and 12 candidate genes were identified.  The procedures utilized appeared to be appropriate to accomplish the study goals.  This manuscript will make a useful contribution to the literature concerning the genetic regulation of growth in swine.  Many changes in wording are needed to improve the grammar and readability of the paper.

Reply 1: Thank you very much for the recognition of our work. We will reply to your comments point by point, and revise the related contents in the manuscript according to your important objective comments.

Comment 2-29: lines 15-16: Change to 'Growth traits are crucial economic traits in the commercial pig industry and have a substantial impact on pig production'. Change in the text: lines 14-15.

line 23:  Change to '...SSC7 were related to average daily gain from 30 to 60 kg...'. Change in the text: lines 21-22.

line 26:  I recommend deleting the word 'dramatically'.  It is rather subjective whether comprehension will be 'dramatically' increased. Change in the text: line 25.

line 37:  Change to '...swine industry and have a...'.   Change in the text: line 36.

line 38:  Change to '...are vital indicators of the growth rate of pigs...'. Change in the text: line 37.

lines 42-44:  Change to ‘Both genetic and non-genetic effects, including pig breed, feeding behavior, and nutrition level, influence growth traits of pigs’. Change in the text: lines 41-43.

line 44:  Change to ‘…pig growth performance has been…’. Change in the text: line 43.

lines 46-47:  Change to ‘…numerous minor genes.  Therefore, improving growth traits through conventional breeding is time-consuming and costly. Change in the text: lines 45-46.

lines 47-49:  Change to ‘Due to the rapid development of molecular markers and the completion of the pig genome sequence, molecular breeding has become an effective way to improve growth traits’. Change in the text: lines 47-48.

lines 50-51:  Change to ‘…total of 2,597 quantitative trait loci (QTL) associated with growth traits have been added to the pig QTL database…’. Change in the text: lines 49-50.

line 60: ‘has’ in place of ‘had’. Change in the text: Line 59.

line 62:  Delete ‘in pigs’.  It is clear that you are talking about pigs. Change in the text: Line 61.

line 67:  Delete ‘besides’. Change in the text: Line 66.

lines 71-72:  Change to ‘…sequencing (WGS) is another genotyping method that has been used in recent years…’. Change in the text: lines 70-71.

line 76:  Delete ‘extremely’. Change in the text: line 74.

line 79:  Change to ‘Abundant SNPs and candidate genes have been identified…’. Change in the text: line 78.

line 82:  Delete ‘successfully’. Change in the text: line 80.

line 85:  Delete ‘pig’. Change in the text: line 83.

line 88:  Delete ‘As we know’. Change in the text: line 86.

line 89:  Change to ‘…are significantly different from Chinese native pigs…’. Change in the text: lines 86-87.

line 98:  Change ‘during’ to ‘from’ in both places. Change in the text: line 96.

line 103:  Change to ‘…this is the first report…’. Change in the text: line 101.

line 104: ‘offer’ in place of ‘offered’. Change in the text: line 102.

line 117:  Change to ‘…and had ad libitum access to feed and water’. Change in the text: line 115.

line 118: Change to ‘…when they weighed an average of 105.25 ± 15.75 kg’. Change in the text: line 116.

line 119: Delete ‘In the present study,’.  It is clear that you are talking about the present study. Change in the text: line 118.

line 123:  Delete ‘Besides’. Change in the text: line 122.

line 126:  Delete ‘the length of’. Change in the text: line 125.

line 129: ‘through the anus’?  Change in the text: lines 126-128.

Reply 2-29: Thank you very much. We strongly agree with your advice. We have revised them in the text according to your advice.

Comment 30-63: line 129:  Delete ‘same’. Change in the text: line 128.

line 130:  Delete ‘among them’. Change in the text: line 128.

line 131: Change ‘sample’ to ‘pig”. Change in the text: line 130.

lines 134-135:  What was the statistical model used to analyze the dependent variables? Change in the text: lines 132-134.

line 137:  Change to ‘…was used for the phenotypic…’. Change in the text: line 136.

line 140: ‘employed’ in place of ‘employ’. Change in the text: line 140.

lines 141, 154, and 182: Delete ‘And’.  Do not begin sentences with ‘And’. Change in the text: lines 141, 154, and 184.

line 157: ‘were’ in place of ‘was’. Change in the text: line 157.

lines 169-170: I believe it would be more correct to state, ‘α and β were fixed effects, while μ and ε were random effects’. Change in the text: lines 171-172.

line 174: Change to ‘…with the trait of interest’. Change in the text: line 176.

lines 174-175:  Change to ‘Based on the number of filtered SNPs (n = 227,921), the threshold…’. Change in the text: line 176.

line 176: Change to ‘…were the genome-wide 10% and 1% significance levels, respectively’. Change in the text: line 178.

line 179: ‘references’ in place of ‘reference’.  You have listed more than one reference. Change in the text: line 181.

line 187:  Change to ‘is shown’. Change in the text: line 195.

line 190:  Change to ‘coefficients of variation’. Change in the text: line 198.

line 191: Change to ‘…7.51, and 8.36%, respectively…’.  The CV is expressed as a percent. Change in the text: line 199.

Table S1:  Change the last column heading to ‘CV, %’. Change in the text: Table S1.

lines 192-193: Change to ‘…crossbred pig populations had a large variation for ADG’.  The CV of 26.58% is large, but I would not call it ‘extraordinary’.  The CVs are an indication of phenotypic variation, not genetic variation. Change in the text: lines 200-201.

lines 193-194:  Change to ‘The frequency distributions of the traits are shown in Figure 1. The nine growth traits appear to conform to the normal…’. Change in the text: lines 201-202.

line 195: The phenotypic correlation coefficients for the nine growth traits are shown in Table 1’. Change in the text: line 203.

Table 1 and in the text: It is not necessary to report the correlation coefficients to 4 decimal places.  Two decimal places are sufficient. Change in the text: Table 1.

line 197:  Delete ‘Besides’. Change in the text: line 204.

line 201: Change ‘during’ to ‘from’ in both places. Change in the text: line 208.

line 204:  Change the title of Table 1 to ‘Phenotypic correlations among nine growth traits in crossbred pigs’. Change in the text: line 211.

lines 205-206: Change ‘during’ to ‘from’ in both places. Change in the text: lines 212-213.

Tables S2 and S3:  Change ‘each accessions’ to ‘each accession’ in the title. Change in the text: Tables S2 and S3.

line 223: ‘normally efficiency’ does not make sense.  Please check it. Change in the text: line 230.

line 227:  What is ‘heter ratio’? Change in the text: line 234.

line 229: Change to ‘…were filtered…’. Change in the text: line 236.

line 233: It does not make sense to say, ‘which indicated that the data was available’.  Please reword. Change in the text: line 240.

lines 240-241:  Which column of Table S6 shows the percentage of phenotypic variation explained? Change in the text: Table S6.

line 241: ‘Among them’?  Among what? Change in the text: line 248.

line 251: Change to ‘…in any genes.  SNP rs322460444 exceeded…’. Change in the text: line 257.

lines 252-253:  Change to ‘…was located 29.6 kb upstream of ATP synthase…’. Change in the text: lines 258-259.

Reply 30-63: Thank you very much. We strongly agree with your advice. We have revised them in the text according to your advice.

Comment 64-95: line 254: Change to ‘…on SSC18 were located 1.4 and 1.5 kb, respectively, upstream of the growth hormone-releasing hormone receptor (GHRHR) gene’. Change in the text: lines 260-261.

line 260: Change to ‘…was located 18 kb upstream of…’. Change in the text: line 266.

line 271:  Change to ‘were located’. Change in the text: line 277.

line 273:  Change to ‘…was located 27.6 kb and 27.8 kb downstream of nudix…’. Change in the text: line 279.

line 277: Change to ‘…on SSC7 were located 6.2 kb downstream of the NHL…’. Change in the text: line 283.

line 279:  Change to ‘…was located 4.4 kb downstream of tripartite motif…’. Change in the text: line 285.

line 280: Change to ‘Two adjacent SNPs on SSC14 were located within the small G protein…’. Change in the text: line 286.

line 289: Delete ‘In this study’.  It is clear that you are presenting results from this study. Change in the text: line 295.

line 290: Change to ‘…was located 20.7 kb upstream of the nuclear factor…’. Change in the text: line 296.

line 291: Delete ‘identified’. Change in the text: line 297.

lines 293-297: Change to ‘…was located 26.5 kb downstream of NCK adaptor protein 2 (NCK2), while the significant SNP (rs705385434) on SSC2 was located 2.8 kb upstream of mastermind like transcriptional coactivator 1 (MAML1). For RC, two adjacent SNPs (rs699438879 and rs712077976) on SSC17 were located within the pleckstrin and Sec7 domain containing the 3…’. Change in the text: lines 299-302.

line 308: Delete ‘The study revealed that’. Change in the text: line 314.

line 314: Change to ‘are shown’. Change in the text: line 319.

Table S7: Delete ‘and newly significant SNPs for growth traits’ from the title of the table. Change in the text: Table S7.

line 320: Change to ‘are shown’. Change in the text: line 326.

line 323: Delete ‘In this study’.  It is clear that you are discussing the results of your study. Change in the text: line 347.

lines 331-332: Avoid using two phrases beginning with ‘which’ in the same sentence. Change in the text: lines 353-356.

line 336: Delete ‘Besides’. Change in the text: line 360.

lines 342-343: The correlation coefficients provide an indication of pleiotropic effects, but they do not really ‘explain’ the pleiotropic effects. Change in the text: lines 365-366.

lines 345-346: Change to ‘…was located 29.6 kb upstream …’. Change in the text: lines 368-369.

line 348: Delete ‘Besides’. Change in the text: line 371.

line 349: Change to ‘…were located 1.4 and 1.5 kb, respectively, upstream of the GHRHR gene…’. Change in the text: line 372.

lines 350 and 354: Change ‘was’ to ‘is’. Change in the text: lines 373, and 377.

line 353: Change to ‘…biological process is that growth…’. Change in the text: line 375.

lines 355-356: ‘sensitively’?  Do you mean ‘sensitivity’? Change in the text: lines 379-380.

lines 357-358: Delete ‘of animals’. Change in the text: line 381.

line 358: Change to ‘…GHRHR regulates…’. Change in the text: line 382.

line 375:  Change ‘functioned’ to ‘function’ as they still function in this fashion. If you say ‘functioned’, it implies that they used to function in this way in the past, but they no longer do so. Change in the text: line 402.

lines 380-383: Change to ‘…since it is developmentally regulated and likely has a significant function during the embryonic and early postnatal skeletal development phases. Given that TRIM55 is functionally related to …’. Change in the text: lines 406-408.

line 384:  Change to ‘…was located 18 kb upstream of the…’. Change in the text: line 410.

line 385: ‘plays’ in place of ‘played’.  Use the present tense. Change in the text: line 411.

Reply 64-95: Thank you very much. We strongly agree with your advice. We have revised them in the text according to your advice. Thank you very much again.

Comment 96-113: line 386: ‘participates’ in place of ‘participated’.  Use the present tense in place of the past tense since it still plays this role.  These types of changes need to be made throughout the entire manuscript. Change in the text: line 412.

lines 390 and 458:  Change to ‘GO’. Change in the text: line 417.

line 412:  Change to ‘…BL is located 27.6 kb and 27.8 kb downstream of NUDT3 and HMGA1, respectively’. Change in the text: line 438.

line 416: Delete ‘only’. Change in the text: line 442.

line 417: Delete ‘besides’. Change in the text: line 443.

line 422: ‘Pigs’ does not need to be capitalized. Change in the text: line 448.

line 430: Change to ‘SNPs on SSC7 were located 8.2 kb downstream of…’. Change in the text: line 456.

lines 431-432: Do not use phrases beginning with ‘which’ twice in the same sentence. Change in the text: lines 457-459.

line 435: Change to ‘…located 4.4 kb downstream of…’. Change in the text: line 461.

line 437: Replace ‘one research’ with ‘one study’. Change in the text: line 463.

line 439: Change to ‘…28 days of age in broilers…’. Change in the text: line 465.

lines 444-445: Change to ‘Only one SNP was significantly associated with CC. The location of the SNP was 20.7 kb upstream of NFATC2’. Change in the text: lines 472-473.

lines 446-447: Delete ‘As everyone knows’. Change in the text: line 474.

 line 453: Change to ‘…the most significant SNP was located 26.5 kb downstream of the NCK2 gene’. Change in the text: line 481.

line 458: Change to ‘…on SSC2 was located 2.8 kb upstream of MAML1…’. Change in the text: line 487.

lines 484-488: Change to ‘Overall, our study provided new evidence that multiple genes are involved in regulating growth traits in pigs. The SNPs and corresponding candidate genes serve as a biological foundation for improving growth and production in swine breeding’. Change in the text: lines 534-537.

line 506: Change to ‘Every effort was made…’. Change in the text: line 554.

Reply 96-113: Thank for your advice. We have revised them in the text according to your advice.

Thank you very much again. 

Reviewer 2 Report

Dear Authors,
congratulations on your work.
I just have a couple of caveats to point out:
- Ethics Statement: you wrote "The ethics committee of Yunnan Agricultural University (YNAU, Kunming, China) approved the entire research" is there an official protocol number?
- Were the animals sacrificed or slaughtered for consumption? Did you notice a phenotypic correlation between subjects expressing candidate genes and improvement in growth traits?

Author Response

Dear reviewer:

We are very grateful for your objective scientific comments. We have carefully revised the manuscript point by point as advised.

Comment 1: Dear Authors, congratulations on your work.

Reply 1: Thank you very much for the recognition of our work and your important scientific advice. I quite agree with your advice. We have made a revision to the manuscript according to your advice. We will answer questions point by point according to the following comments. Thank you very much again.

Comment 2: I just have a couple of caveats to point out:

- Ethics Statement: you wrote "The ethics committee of Yunnan Agricultural University (YNAU, Kunming, China) approved the entire research" is there an official protocol number?

Reply 2: Thank you very much. The ethics committee of Yunnan Agricultural University (YNAU, Kunming, China) approved the entire research, but we have don’t an official protocol number because Yunnan Agricultural University won't have an ethical approval number until 2020. Our experiment was conducted before 2020. Thank for your advice again.

Comment 3: - Were the animals sacrificed or slaughtered for consumption?

Reply 3: Thank you very much. The animals in the experiment have been sacrificed for carcass and meat quality traits measurement.

Comment 4: Did you notice a phenotypic correlation between subjects expressing candidate genes and improvement in growth traits?

Reply 4: Thank you very much. At present, we have not conducted association verification tests on candidate genes and growth traits. We will carry out these tests in the future. In the study, we added the association analysis between genotyping of significant SNPs and related growth traits. Chang in the text: lines 190-195, 332-349, Table 2, 392-393, 421-422, 472-473, and 492-493.

Thank you very much again.

Reviewer 3 Report

Authors have performed GWAS for growth traits, one of the most important traits in the pig industry. My first concern is about choosing the SLAF-seq instead of the genotyping technology. I could not see a clear benefit of this technology. My most concern is about the sample size, it is too small for the GWAS study. Did the authors check the power of GWAS?

Line 24-36: Perhaps better to add growth and body conformation traits as some traits are not growth traits.

Line 41: might change feed intake capacity to feed efficiency.

Line 42-44: Providing the references.

Line 60-61: Add more references regarding GWAS for ADGs and other fat-related traits.

Line 61-75: The authors focused much on the studies based on Chinese pig breeds, the authors might extend the introduction to GWAS for pigs in other countries.

Line 72-73: I do not think so, given the cost of medium-density chips has been significantly reduced.

Line 81-83: Are any overlapping in the animals used in the current study with the previous one? Please clarify.

Why did the authors divide the ADGs into two different traits?

Line 156-157: add more details about the parameters used for sequence processing and

variant callings

Line 171: No need to use an abbreviation of BC.

Table 1: Using 2 digitals is enough

The authors should also discuss why no correlations between ADG30-60 and ADG 60-100.

Author Response

Dear reviewer:

We are very grateful for your objective scientific comments. We have carefully revised the manuscript point by point as advised.

Comment 1: Authors have performed GWAS for growth traits, one of the most important traits in the pig industry.

Reply 1: Thank you very much for your important scientific advice. We have made a revision to the manuscript according to your advice. We will answer questions point by point according to the following comments. Thank you very much again.

Comment 2: My first concern is about choosing the SLAF-seq instead of the genotyping technology. I could not see a clear benefit of this technology.

Reply 2: specific-locus amplified fragment sequencing (SLAF-seq) based on a reduced representation library and high-throughput sequencing. This technique has several distinguishing characteristics: deep sequencing, reduced sequencing costs, optimized marker efficiency, and applicability to large populations [1]. Based on the reference genome, a pre-experiment of SLAF-seq is carried out to select the appropriate combination of enzyme digestion, so as to produce a sufficient number of tags covering the whole genome and effectively avoid repeated sequences. In the study, the pre-experiment of SLAF-seq was based on the pig reference genome to screen the appropriate enzyme digestion combination (RsaI and HaeIII), which ensured the number and depth of SLAF tags covering the whole genome (Table S3 and Table S5).

Comment 3: My most concern is about the sample size, it is too small for the GWAS study. Did the authors check the power of GWAS?

Reply 3: In the study, we used a four-way crossbred pig population, in which Landrace, Yorkshire, and Duroc were used as hybrid males and Saba pigs as hybrid females. The crossbred population showed great differences in growth traits and more genetic variation. Through the QQ plot, we found that the results of GWAS were satisfactory.

Furthermore, we added the association analysis between genotyping of significant SNPs and related growth traits. The results showed that three genotypes of these SNPs presented significant differences (p<0.05) in the corresponding growth traits.

Comment 4-6: Line 24-36: Perhaps better to add growth and body conformation traits as some traits are not growth traits. Change in the text: line 33.

Line 41: might change feed intake capacity to feed efficiency. Change in the text: line 40.

Line 42-44: Providing the references. Change in the text: lines 43, 586-594.

Reply 4-6: Thank for your advice. We have revised them in the text according to your advice. Thank you very much again.

 Comment 7: Line 60-61: Add more references regarding GWAS for ADGs and other fat-related traits.

Reply 7: Thank you very much, we have added several references regarding GWAS for ADGs and other fat-related traits. Change in the text: lines 60-61, 624-635.

Comment 8: Line 61-75: The authors focused much on the studies based on Chinese pig breeds, the authors might extend the introduction to GWAS for pigs in other countries.

Reply 8: thank you very much, we have extended the introduction to GWAS for pigs in other countries. Change in the text: lines 65-67.

Comment 9: Line 72-73: I do not think so, given the cost of medium-density chips has been significantly reduced.

Reply 9: thank you very much, Line 72-73: GWAS for Sus Scrofa with large populations based on WGS is still prohibitively expensive at present. WGS is an abbreviation for whole-genome sequencing. Additionally, we strongly agree with the cost of medium-density chips has been significantly reduced, but GWAS based on chips can’t detect novel SNP loci. Therefore, SLAF-seq used in the GWAS study for large populations was a better choice in the current higher sequencing cost.

Comment 10: Line 81-83: Are any overlapping in the animals used in the current study with the previous one? Please clarify.

Reply 10: Thank you very much, we used the same crossbred pigs in the two studies.

Comment 11: Why did the authors divide the ADGs into two different traits?

Reply 11: Thank you very much. Growing pigs (30-60kg) and finishing pigs (60-100kg) have different developmental characteristics. Pigs in the early growth period are mainly characterized by the growth and development of muscle and bone, while Pigs in the late growth period are mainly characterized by fat deposition

Comment 12: The authors should also discuss why no correlations between ADG30-60 and ADG 60-100.

Reply 12: Thank for your advice. We strongly agree with you. We have added the discuss why no correlations between ADG30-60 and ADG 60-100. Change in the text: lines 515-535.

Comment 13: Line 156-157: add more details about the parameters used for sequence processing and variant callings.

Reply 13: Variation detection: GATK v3.8

  1. Local realignments: the regions for realignments were excavated using “realignertargetcreator” in the GATK tool, and were corrected using indelrealigner. 2. Mutation detection-haplotypecaller: gvcf generation using haplotypecaller (haplotype local assembly algorithm). Main parameters: java   -jar GenomeAnalysisTK.jar -T HaplotypeCaller -R ref.fa --emitRefConfidence GVCF --variant_index_type LINEAR --sample_ploidy 2 -o *.g.vcf  -I *..bam. 3. Gvcf merge-combinegvcfs: java -jar GenomeAnalysisTK.jar -T CombineGVCFs -R ref.fa. 4. Genotyping-genotypegvcfs: java -jar GenomeAnalysisTK.jar -T GenotypeGVCFs -R ref.fa.

A more detailed description of GATK [2] and SAMtools [3] was presented in the original articles.

Chang in the text: lines 160-162.

Comment 14-15: Line 171: No need to use an abbreviation of BC. Chang in the text: lines 176.

Table 1: Using 2 digitals is enough. Chang in the text: Table 1.

Reply 14-15: Thank you very much. We have revised them in the text according to your advice.

Thank you very much again.

 References

  1. Sun, X.; Liu, D.; Zhang, X.; Li, W.; Liu, H.; Hong, W.; Jiang, C.; Guan, N.; Ma, C.; Zeng, H.; et al. SLAF-Seq: An Efficient Method of Large-Scale De Novo SNP Discovery and Genotyping Using High-Throughput Sequencing. PLoS ONE 2013, 8, e58700. https://doi.org/10.1371/journal.pone.0058700.
  2. McKenna, A.; Hanna, M.; Banks, E.; Sivachenko, A.; Cibulskis, K.; Kernytsky, A.; Garimella, K.; Altshuler, D.; Gabriel, S.; Daly, M.; et al. The Genome Analysis Toolkit: A MapReduce Framework for Analyzing Next-Generation DNA Sequencing Data. Genome Res. 2010, 20, 1297-1303. https://doi.org/10.1101/gr.107524.110.
  3. Li, H.; Handsaker, B.; Wysoker, A.; Fennell, T.; Ruan, J.; Homer, N.; Marth, G.; Abecasis, G.; Durbin, R.; Genome Project Data Processing, S. The Sequence Alignment/Map Format and Samtools. Bioinformatics 2009, 25, 2078-2079. https://doi.org/10.1093/bioinformatics/btp352.

Round 2

Reviewer 3 Report

The authors have responded to my comments. The manuscript has improved.  For discussion about the sample size, the authors might consider this response as a limitation of the manuscript to ensure the reader is aware of the difficulty of collecting the phenotypes/genotypes in the current manuscript. 

Since the authors mentioned that "Reply 10: Thank you very much, we used the same crossbred pigs in the two studies.", the authors should clarify in the revised manuscript which are overlapping and which are novel data in the current study. 

For splitting ADG into two different periods, I am not convinced with the explanation. At least the authors should try to make ADG 30-90 to make it compatible with previous studies.  The difference in the results between the two periods as in the current manuscript might just be due to the fact the sample size is too small. From a biological perspective, it is very surprising that there is no correlation between the two ADGs, did the authors have a chance to check the genetic correlation or some supporting evidence? 

Author Response

Dear reviewer:

We are very grateful for your objective scientific comments. We have carefully revised the manuscript point by point as advised.

Comment 1: The authors have responded to my comments. The manuscript has improved.

Reply 1: Thank you very much for your important scientific advice. We have made a revision to the manuscript according to your advice. We will answer questions point by point according to the following comments. Thank you very much again.

Comment 2: For discussion about the sample size, the authors might consider this response as a limitation of the manuscript to ensure the reader is aware of the difficulty of collecting the phenotypes/genotypes in the current manuscript.

Reply 2: Thank for your advice. We have revised them in the text according to your advice. Thank you very much again.

Change in the text: lines 11-13 of “2.2 Animals, Phenotypic Collection and Statistical Analysis” of the “Materials and Methods” section;

Comment 3: Since the authors mentioned that "Reply 10: Thank you very much, we used the same crossbred pigs in the two studies.", the authors should clarify in the revised manuscript which are overlapping and which are novel data in the current study.

Reply 3: Thank for your advice. We have revised them in the text according to your advice. Thank you very much again.

Change in the text: line 1 of “2.2. Animals, Phenotypic Collection and Statistical Analysis” of the “Materials and Methods” section;

Lines 1-2 of “2.3. SLAF Library Construction and High-throughput Sequencing” of the “Materials and Methods” section;

Lines 1-2 of “3.2. Identification of SLAFs and SNPs” of the “Results” section;

Comment 4: For splitting ADG into two different periods, I am not convinced with the explanation. At least the authors should try to make ADG 30-90 to make it compatible with previous studies. The difference in the results between the two periods as in the current manuscript might just be due to the fact the sample size is too small. From a biological perspective, it is very surprising that there is no correlation between the two ADGs, did the authors have a chance to check the genetic correlation or some supporting evidence?

Reply 4-6: Thank for your advice. We have added phenotypic data and GWAS results of ADG30-100 trait and checked the genetic correlation between three ADG traits according your advice. We found that ADG30-60 and ADG60-100 had weak negative genetic correlation.

Change in the text: lines 65-72 of “4.2. Candidate Genes for LBT, ADG30-60, ADG60-100 and ADG30-100” of the “Discussion” section;

Lines 1-29 of “4.5. Comparison between ADG30-60 and ADG60-100” of the “Discussion” section;

Thank you very much again.
